# 1 Heteronuclear and Homonuclear Finite Pulse Radio

| 2 | Frequency | Driven | Recoupling |
|---|-----------|--------|------------|
|---|-----------|--------|------------|

- Authors: Evgeny Nimerovsky\*, Kai Xue, Kumar Tekwani Movellan & Loren B. Andreas\*
- Affiliations:
- Department of NMR based Structural Biology, Max Planck Institute for Biophysical Chemistry, Am
- Fassberg 11, Göttingen, Germany
- \*Corresponding authors: land@nmr.mpibpc.mpg.de ORCID: 0000-0003-3216-9065 and
- evni@nmr.mpibpc.mpg.de
- Abstract

Homonuclear finite-pulse radio frequency driven recoupling (fp-RFDR) has been broadly used in 11 multi-dimensional magic-angle spinning (MAS) solid-state NMR experiments over the past 20 years. The 12 theoretical and the simulated descriptions of this method were presented during that time, resulting in an 13 understanding of the influence of chemical shift offset, finite pulse effects, and dipolar truncation. Here we present an operator analysis of both heteronuclear and homonuclear fp-RFDR. By numerical 14 15 simulation, we show which operators are involved in the longitudinal exchange for both heteronuclear 16 and the well-known homonuclear sequences. This results in a better understanding of the influence of 17 phase cycling of the fp-RFDR pulses, which is typically a variant of XY cycling. We investigate the heteronuclear and homonuclear fp-RFDR signals and evolution of the operators through the fp-RFDR 18 block. We show the convergence of the evolutions of the heteronuclear and homonuclear fp-RFDR 19 20 signals at even numbers of rotor periods and completely different evolution between them. We demonstrate heteronuclear <sup>1</sup>H-<sup>13</sup>C and <sup>1</sup>H-<sup>15</sup>N fp-RFDR magnetization transfer using a microcrystalline 21 22 SH3 sample at 100 kHz MAS.

23 Keywords: Magic Angle Spinning NMR, homonuclear and heteronuclear fpRFDR, the operator analysis

#### 24 Introduction

Solid-state magic-angle spinning NMR spectroscopy is used to obtain atomic resolution physical 26 and chemical knowledge about the investigated sample. One of the abilities of the NMR experiments is to determine distance between a pair of spin 1/2 nuclei via recoupling the homonuclear(A. E. Bennett et al. 27 28 1992; Ok et al. 1992; Z. Zhang et al. 2020; Gelenter, Dregni, and Hong 2020; Takegoshi, Nakamura, and Terao 2001; Szeverenyi, Sullivan, and Maciel 1982; Hou, Yan, et al. 2011; Hou et al. 2013; Carravetta et 29 al. 2000; Andrew E. Bennett et al. 1998) or heteronuclear(Gelenter, Dregni, and Hong 2020; T. Gullion 30 31 and Schaefer 1989; Jaroniec, Filip, and Griffin 2002; Hing, Vega, and Schaefer 1992; Hartmann and 32 Hahn 1962; Rovnyak 2008; Metz, Wu, and Smith 1994; Hediger et al. 1994; Hou, Byeon, et al. 2011; Brinkmann and Levitt 2001; Gelenter and Hong 2018; Z. Zhang, Chen, and Yang 2016) dipolar 33 34 interactions. The homonuclear fp-RFDR sequence is successfully applied for the qualitative and quantitative determinations of the dipolar spin correlations in materials(Saalwächter 2013; Messinger et 35 36 al. 2015; Fritz et al. 2019; Roos, Mandala, and Hong 2018; Nishiyama et al. 2014; Wong et al. 2020; 37 Hellwagner et al. 2018; Pandey and Nishiyama 2018) and biomolecular samples(Zheng, Qiang, and 38 Weliky 2007; Tang, Berthold, and Rienstra 2011; Shen et al. 2012; Pandey et al. 2014; Grohe et al. 2019; Andreas et al. 2015; Petkova et al. 2002; Aucoin et al. 2009; Zinke et al. 2018; R. Zhang, Mroue, and 39 40 Ramamoorthy 2017; Zhou et al. 2012; Jain et al. 2017; Colvin et al. 2015; Shi et al. 2015; Daskalov et al. 41 2020). 42 These applications depend on a firm quantum mechanical foundation. One of the theoretical tools to investigate the influence of radio frequency (RF) pulse sequences on the spin system is Average 43 44 Hamiltonian Theory(Haeberlen and Waugh 1968; Maricq 1982) (AHT). The two necessary conditions for

application of the AHT are(Ernst, Bodenhausen, and Wokaun 1987):

1. The total Hamiltonian has to be periodic;

2

| 47 | 2. The stroboscopic measurements are synchronized with the period, or cycle time, of the total                 |
|----|----------------------------------------------------------------------------------------------------------------|
| 48 | Hamiltonian.                                                                                                   |
| 49 | When these conditions are fulfilled, the time-dependent Hamiltonian, evaluated at multiples of the cycle       |
| 50 | time, can be replaced by the sum of the time-independent multiple order averaging terms(Ernst,                 |
| 51 | Bodenhausen, and Wokaun 1987).                                                                                 |
| 52 | AHT simplifies quantum calculations, especially in cases when complex multiple-pulse                           |
| 53 | sequences are used. It can explain the selectivity of the pulse sequence, meaning to find the experimental     |
| 54 | conditions under which the desired interactions are recoupled and undesired decoupled. However, AHT            |
| 55 | can predict the state of the spin system at specific time points only and not the paths by which the spin      |
| 56 | system is evolves during the period when rf pulses are given. Another successful method, Floquet               |
| 57 | Theory(Levante et al. 1995; Scholz, van Beek, and Ernst 2010), allows to consider the Hamiltonian at any       |
| 58 | point of time. However, such analysis is complicated with a transformation to infinity-dimensional             |
| 59 | Hilbert space(Levante et al. 1995).                                                                            |
| 60 | Homonuclear transfer of the magnetization via longitudinal exchange occurs with a rotor-                       |
| 61 | synchronized train of $\pi$ -pulses, with one pulse each rotor period. The method is called radio-frequency    |
| 62 | driven recoupling(A. E. Bennett et al. 1992) (RFDR), or simple excitation for the dephasing of rotational-     |
| 63 | echo amplitudes(Terry Gullion and Vega 1992). This sequence has two different AHT descriptions of the          |
| 64 | recoupling of homonuclear dipolar interactions, depending on the experimental conditions(Ok et al. 1992;       |
| 65 | Ishii 2001).                                                                                                   |
| 66 | In the first case, delta $\pi$ -pulses are assumed. The efficiency to recouple homonuclear dipolar             |
| 67 | interaction is linked with the difference between isotropic chemical shifts of the dipolar linked spins, $S_k$ |
| 68 | and $S_l$ (A. E. Bennett et al. 1992; Terry Gullion and Vega 1992; Andrew E. Bennett et al. 1998). The         |

evolution of the spin system at specific time points is described with a flip-flop part of zero-quantum

3

- dipolar Hamiltonian:  $I_k^+ I_l^- + I_k^- I_l^+$  (A. E. Bennett et al. 1992; Andrew E. Bennett et al. 1998; Nielsen et
- al. 1994; Ok et al. 1992; Bayro et al. 2009; Sodickson et al. 1993; Straasø et al. 2016).
- For the second theoretical description, finite  $\pi$ -pulses are considered (fpRFDR)(A. E. Bennett et
- al. 1992; Ishii 2001; Nishiyama, Zhang, and Ramamoorthy 2014; R. Zhang et al. 2015; Brinkmann,
- Schmedt auf der Günne, and Levitt 2002; Ji et al. 2020). The efficiency of recoupling of the homonuclear
- dipolar interaction is directly linked with a duty factor(Ishii 2001) the ratio between the width of  $\pi$ -
- pulse and the width of the rotor period. AHT predicts restoring of the whole zero-quantum dipolar
- Hamiltonian,  $3I_{kz}I_{lz} \bar{I}_k\bar{I}_l$ , under fast and ultra-fast MAS rates. (Ishii 2001)
- Both these theoretical descriptions consider the same experiment with the same phase cycling, traditionally XY8 (Terry Gullion, Baker, and Conradi 1990). Although the influence of the different phase cycling schemes was investigated in different articles(Ok et al. 1992; Nishiyama, Zhang, and Ramamoorthy 2014; R. Zhang et al. 2015; Ji et al. 2020), the main conclusion to the contribution from phase cycling to the transfer of the RFDR signal was a reduction of influence from resonance offsets and pulse errors(A. E. Bennett et al. 1992; Ishii 2001).
- In this article we investigate both heteronuclear and homonuclear fpRFDR experiments using 84 numerical tools to track the system at any arbitrary time. Using the simulated and the theoretical analysis 85 86 we show that for fpRFDR experiments the typical phase cycling, XY(Ishii 2001; Nishiyama, Zhang, and 87 Ramamoorthy 2014; R. Zhang et al. 2015; Hellwagner et al. 2018), plays a crucial role in the transfer of 88 magnetization between a pair of spins. Under fast and ultra-fast MAS rates the heteronuclear and 89 homonuclear fpRFDR experiments can be described with the same model Hamiltonian, but only at 90 increments of the rotor period. The evolutions of the operators, however, are completely different for 91 heteronuclear and homonuclear cases between these points. For the experimental demonstrations we perform heteronuclear 1D  ${}^{1}H{-}{{}^{13}C}$  and  ${}^{1}H{-}{{}^{15}N}$  fp-RFDR experiments using  $\alpha$ -PET (Movellan et al. 92 93 2019) labeled SH3 at 100 kHz MAS.

#### 94 Theory

- The fp-RFDR sequence consists of a train of  $\pi$ -pulses every one rotor period (Fig.1a). The length
- of the repeated block is defined by the phase cycling: XYn (n=4, 8, 16, 32), resulting in a time of  $nT_R$
- (Ishii 2001). Measurements are, in the simplest case, restricted to occur every  $nT_R$ . In our simulations as
- well as in the experiments we used XY8 phase cycling.

To evaluate the operator,  $\hat{A}$ , between the time points  $t_i$  and  $t_i + t_x$  we first have to solve the von

Neumann equation ( $\hbar = 1$ ):

$$\frac{d\rho}{dt} = -i[H(t), \rho(t)], \text{ Eq. (1.1)}$$

where H(t) is a Hamiltonian of the spin system and  $\rho(t)$  is a density matrix. The formal solution of the

$$\rho(t_i + t_x) = \hat{T}exp\left\{-i\int_{t_i}^{t_i + t_x} dt \, H(t)\right\}\rho(t_i)\hat{T}exp\left\{-i\int_{t_i}^{t_i + t_x} dt \, H(t)\right\}, \text{ Eq. (1.2)}$$

where  $\hat{T}$  is a Dyson Operator.

The evaluated operator,  $\hat{A}(t_i + t_x)$  is:

$$\hat{A}(t_i + t_x) = Tr\{\hat{A}\rho(t_i + t_x)\}$$
 Eq. (1.3)

One of the possibilities to deal with a Dyson Operator in Eq. (1.2), in order to propagate forward 109 in time from the point  $t_i$  to  $t_i + t_x$ , is to split one propagator into a product of *N* propagators(Nimerovsky 110 and Goldbourt 2012):

$$\widehat{T}exp\left\{-i\int_{t_i}^{t_i+t_x} dt \, H(t)\right\} = \lim_{N \to \omega} \prod_{k=1}^N exp\left\{-i\int_{t_i+t_x-\Delta_{x,i}(k-1)}^{t_i+t_x-\Delta_{x,i}(k-1)} dt \, H(t)\right\}, \ \Delta_{x,i} = t_x/N \quad \text{Eq. (1.4)}.$$

It allows to omit the Dyson Operator and perform the simulations correctly. This is the way that

calculations are performed in the popular SIMPSON software(Bak, Rasmussen, and Nielsen 2000). The

main difference in our implementation of numerical evolutions with respect to SIMSPON calculations is

to use analytical integrations rather than numerical integrations, which significantly reduces the computer time.

Each of the rotor periods of the fp-RFDR sequence can be divided into two parts with the lengths 118 of  $t_p$  (defined with the length of  $\pi$ -pulse and  $T_R - t_p$  (the delay, Fig. 1a). The numerical calculations split each of these two parts into N subparts with the lengths  $t_p/N$  and  $(T_R - t_p)/N$ , respectively. Fig. 1b shows 119 120 the transferred fp-RFDR signals for  $I_3$  spin system under different values of N. Solid lines represent the 121 transferred  $I_2 \rightarrow I_3$  signal between a weakly coupled dipolar pair with 66 Hz dipolar interaction, whereas 122 the dotted lines represent the  $I_2 \rightarrow I_2$  signal. With an increase in the value of N, the simulated signals 123 converge and under N=16 (red lines) and N=32 (black lines) the signals coincide. It means that under 124  $N \ge 16$ , the simulations provide the correct evolution of the spin system. In all numerical calculations we 125 used N=32.