# Peer review of "Heteronuclear and Homonuclear Finite Pulse Radio"

_Magnetic Resonance, 2020_

## Referee Comment (RC1) · Anonymous Referee #1 · 4 Dec 2020

The paper by Nimerovsky and coworker describes fp RFDR applied to homonuclear and heteronuclear coupled dipolar spin systems.

My first impression after reading the manuscript was, that there is no clear focus and clear message of the paper draft. Is the main message a theoretical framework to analyze such sequences (in this case, it is not clear to me what is new) or is it the use of simultaneous RFDR-like sequences on heteronuclear spins systems to achieve heteronuclear polarization transfer. The second point is that the theory part and the first part of the simulations part seems to be very much detached and unconnected to the main part of the paper. I must admit that the manuscript left me a bit at a loss how things are connected and what the authors want to tell.

The theory part is quite lengthy and discusses the details of time slicing used in nu-

[Figure]

Interactive
comment

merical simulations. There is nothing new in this part and it could be left away with any loss of information. But since it is here, I was wondering about two things: (1) Why is the pulse and the remaining rotor period divided into an equal number of time slices. Typically, one would implement time slicing such that the rotor period is divided into a certain number of slices and the pulse occupies whatever time it takes. Here it says explicitly: "The numerical calculations split each of these two parts into N sub parts with the lengths tp/N and (TR – tp)/N, respectively." If SIMPSON was used for the simulations, this is most likely not correct. And why would one do this and have a higher sampling rate during the pulse? (2) Eq. (2.4) is only correct if the Hamiltonian H(t) commutes with itself over the time period Delta of the integration. This is however not how numerical simulations are typically implemented when doing time slicing. Usually, it is assumed that the Hamiltonian is constant over the time interval used for the calculations and I am pretty sure that this is how SIMPSON implements it. I think the statements and the equation at the end of page 5 are wrong.

In the simulation part, first different approximations for the Hamiltonian are used to understand which part of it is responsible for the polarization transfer under different conditions. The Hamiltonian of Eq. 2.1 makes sense (full rotating-frame truncated dipolar Hamiltonian) but it would be nice to rewrite the second part with 2zz instead of 3zz-zz but the ones selected in Eq. 2.2 do not really make sense or are wrong. The zz only Hamiltonian should have a coefficient of 1 and not 1.5! The second one is a J coupling Hamiltonian which makes no sense in this context while the third one is the ZQ part which is sensible. If these are indeed the Hamiltonians used for the simulations, I would put a big question mark on the conclusions presented in Figs. 3-5. This is implied in the figure labeling but of course I cannot judge what was used in the simulations. On page 11, the heteronuclear Hamiltonian is discussed: "The main difference to the heteronuclear full dipolar Hamiltonian with a homonuclear model Hamiltonian (Eq. 2.1a) is a factor of 1.5." This is clearly not correct. The heteronuclear Hamiltonian has just the zz terms and is exactly the same as a homonuclear (weak-coupling) dipolar Hamiltonian. There is no factor of 1.5 difference between the two.

Where we can find a factor of 1.5 is in the transition frequencies after diagonalisation. All these statements that are clearly wrong make me worry about the results presented in this paper. There is another statement that worries me (page 13): "For negligible offset differences with respect to the MAS rate, the evolution of the operators of IS and I2 spin systems are the same at specific time points." What are negligible offset differences in a heteronuclear spin system when the two spins are in different rotating frames? And further on: "With increase of offset difference the IS spin system passes through specific rotor resonance condition (the difference between offsets equal to half of the MAS rate), under which the transfer does not occur. For the I2 spin system the velocity of the transfer increases with increased offset difference." This cannot be. Either the authors mix up homo- and heteronuclear spin systems or this is plain wrong. There are no MAS resonance conditions in heteronuclear spin systems that depend on any offsets (what offset differences are there?).

Even if some of the problems are just a mix up, all of these mistakes make me wonder how carefulthe rest of the work is done. I have not and will not look at the rest of the manuscript before all these problems are corrected or explained.

Some minor remarks that should be cleaned up:

(1) The introduction has a long list of references to homonuclear and heteronuclear recoupling experiments that is clearly incomplete. Instead of such an incomplete list, it would be much better to have a reference to a review. There is for example a recent review about dipolar resoupling by N.C. Nielsen, L.A. Strassø, A.B. Nielsen, (Top. Curr. Chem. 306 (2012) 1–45) as well as older reviews on this topic. I think it would be more appropriate as a reference than an incomplete list.

(2) page 5 line 99/100: Liovuille-von Neumann equation is much more common. Why hbar=1? Usually the Hamiltonian in NMR is in frequency units and then there is no hbar in front of the commutator.

(3) Equation numbering should be continuous from (1) to (N) without chapter subdivisions.

---

## Author Comment (AC1) · 7 Dec 2020

We are extremely grateful to the reviewer for her/his time and effort in reviewing the manuscript. We apologize for the unclear description of the main ideas of the article. It is true that there is not a single focus to this article, however, we believe it does not make any sense to split this into multiple parts. We hope that the following explanations below will convince the reviewer that what he/she refers to as 'wrong' is in fact correct but lacked sufficient explanation. While we could likely have published an article on heteronuclear RFDR alone, we feel that the more complete article will better serve the solid-state NMR community.

The paper by Nimerovsky and coworker describes fp RFDR applied to homonuclear and heteronuclear coupled dipolar spin systems.
My first impression after reading the manuscript was, that there is no clear focus and clear message of the paper draft. Is the main message a theoretical framework to analyze such sequences (in this case, it is not clear to me what is new)…

The main goal of the article is to show a theoretical interpretation of the fp-RFDR sequence and the mechanisms with which the fp-RFDR signal is transferred between the spins. We analyze the fp-RFDR sequence using numerical code written by us. The main conclusions are:

1. The heteronuclear and homonuclear transferred fp-RFDR signals can be obtained with the same exactly sequence. For the homonuclear spin system the transfer is described by $I_{zk}$ $\xrightarrow{fp-RFDR} I_{zp}$, whereas for the heteronuclear spin system the transfer is described by $I_z$ $\xrightarrow{fp-RFDR} S_z$. The heteronuclear fp-RFDR transfer has not been described before.

2. The heteronuclear and homonuclear transferred signals coincide completely at the time points of whole number of rotor periods, if the ratio between values of the heteronuclear ($\nu_{D,IS}$) and homonuclear ($\nu_{D,IkIp}$) dipolar coupling constants are 1.5 and the differences between offset values of the spins ($\Omega_{IS}$ for heteronuclear case and $\Omega_{IkIp}$ for homonuclear case) are significantly smaller with respect to the used MAS rate ($\nu_R$): $\frac{\Omega_{IS}}{\nu_R} \ll 1$ and $\frac{\Omega_{IkIp}}{\nu_R} \ll 1$. These conditions are firstly demonstrated in this article.

3. The heteronuclear and homonuclear transferred signals can be described with the same model Hamiltonian, if the conditions, $\frac{\Omega_{IS}}{\nu_R} \ll 1$ and $\frac{\Omega_{IkIp}}{\nu_R} \ll 1$, are met and the measurements occurs every whole number of rotor periods. It does not mean that this model Hamiltonian describes the evolutions of the heteronuclear and homonuclear spin systems correctly at every time point of the transfer, which is shown in the article. The model Hamiltonians are indeed not correct for arbitrary conditions, but are useful under the conditions described in the article. Their use is firstly demonstrated in this article.

4. XY phase cycling plays a crucial role in the fp-RFDR transfer. With XX phase cycling the fp-RFDR transfer is always zero at the time points of whole number of rotor periods for both homonuclear and heteronuclear spin system (under condition that $\Omega_{IS} = 0$ and $\Omega_{IkIp} = 0$). We show it with the numerical simulations and provide the theoretical proof for $I_2$ homonuclear spin system using fictitious half spin formalism[1]. This conclusion is opposite with respect to the traditional statement in which the role of XY phase cycling was mentioned as the secondary[2] and fp-RFDR transfer could be achieved with any phase cycling[3]. The crucial role of XY phase cycling in the fp-RFDR transfer is firstly demonstrated in the article.

5. The presence of the offset difference not only has an influence on the efficiency of the transfer, it changes the mechanism of the transfer. It was shown with the model Hamiltonians,

since under different simulated conditions we used different model Hamiltonians. This idea is consistent with AHT literature that demonstrates 2 mechanisms for RFDR.

6. Investigating the evolution of the $IS$ and $I_2$ spin system during two rotor periods we show the paths with which the heteronuclear and homonuclear fp-RFDR signals are transferred. The fp-RFDR signals for both spin systems are manly transferred via mix of Double and Zero Quantum terms. If for $IS$ spin system, only one path is used for transfer the signal: $I_z \rightarrow I_x S_y \rightarrow S_z$, the homonuclear transferred signal uses 64 paths. Despite such huge difference of the evolutions during these two rotor periods, the amplitudes of the operators in the end of two rotor periods coincide completely. The next Fig. shows the schematic illustration of these paths (this Fig. was prepared as a graphical abstract). These paths are firstly demonstrated in this article.

**Heteronuclear fpRFDR path**   **Homonuclear fpRFDR paths**

[Figure]

The fp-RFDR paths for heteronuclear IS (left) and homonuclear $I_2$ (right) spin systems during first two rotor periods.

…or is it the use of simultaneous RFDR-like sequences on heteronuclear spins systems to achieve heteronuclear polarization transfer.

The fp-RFDR sequence was used without any modifications. For the theoretical description only the internal Hamiltonian indicates if the homonuclear or the heteronuclear spin system is considered. With respect to rf-field Hamiltonian, there is no any difference if we apply a train of $\pi$-pulses on N homonuclear ½ spins or we apply a train of $\pi$-pulses on N heteronuclear ½ spins.

The second point is that the theory part and the first part of the simulations part seems to be very much detached and unconnected to the main part of the paper. I must admit that the manuscript left me a bit at a loss how things are connected and what the authors want to tell.

In the "Theory" section we explain how the numerical simulations were performed. We used the equation of motion (the Liovuille-von Neumann equation, Eq. 1.1 in the article, page 5) to calculate the evolutions of the spin operators (Eq. 1.3 in the article, page 5). To deal with the Dyson operator, we split two propagators between 0 and $t_p$ (during the pulse) and between $t_p$ and $T_R$ (during the delay) on N sub-propagators each. After the analytical integration each of these sub-propagators were diagonalized. This idea was firstly implemented on $IS$ spin system, where S spin was 3/2 spin with a large quadrupolar coupling interaction[4]. We will add a sentence to the revised article that: "*This section describes how our numerical simulations were performed.*".

The theory part is quite lengthy and discusses the details of time slicing used in numerical simulations. There is nothing new in this part and it could be left away with any loss of information.

We agree that this section shows nothing new. We emphasize it in that section, page 5, lines 112-113: "*This is the way that calculations are performed in the popular SIMPSON software*". However, we still have to explain how our numerical calculations were performed. In particular, most journals encourage providing a level of detail that allows the reproduction of results independent of other published literature.

But since it is here, I was wondering about two things: (1) Why is the pulse and the remaining rotor period divided into an equal number of time slices. Typically, one would implement time slicing such that the rotor period is divided into a certain number of slices and the pulse occupies whatever time it takes. Here it says explicitly: "The numerical calculations split each of these two parts into N sub parts with the lengths tp/N and (TR – tp)/N, respectively." If SIMPSON was used for the simulations, this is most likely not correct. And why would one do this and have a higher sampling rate during the pulse?

The numerical simulations were performed with our own written programs. For some simulations we used ultra-fast MAS of 90 kHz (page 10, Fig. 4 and page 29 Fig. A1c-d). In that cases the duty factor was significantly increased. Therefore, we used the same N value for both these periods of time to keep it as a constant under different simulated conditions. We agree that the slicing could be done in more sophisticated way. However, the result would be the same.

(2) Eq. (2.4) is only correct if the Hamiltonian H(t) commutes with itself over the time period Delta of the integration. This is however not how numerical simulations are typically implemented when doing time slicing. Usually, it is assumed that the Hamiltonian is constant over the time interval used for the calculations and I am pretty sure that this is how SIMPSON implements it. I think the statements and the equation at the end of page 5 are wrong.

We assume that reviewer refers to Eq.(1.4). We agree that in SIMPSON simulations the Hamiltonian is assumed as a constant over the time interval, which is a kind of numerical integration. It was mentioned in the article (page 5, lines 113-116): "*The main difference in our implementation of numerical evolutions with respect to SIMSPON calculations is to use analytical integrations over a small slice rather than numerical integrations...*".

However we agree that the last part of this sentence is not correct for this specific situation (…", *which significantly reduces the computer time*."), and it is removed.

In the simulation part, first different approximations for the Hamiltonian are used to understand which part of it is responsible for the polarization transfer under different conditions. The Hamiltonian of Eq. 2.1 makes sense (full rotating-frame truncated dipolar Hamiltonian) but it would be nice to rewrite the second part with 2zz instead of 3zz-zz but the ones selected in Eq. 2.2 do not really make sense or are wrong. The zz only Hamiltonian should have a coefficient of 1 and not 1.5! The second one is a J coupling Hamiltonian which makes no sense in this context while the third one is the ZQ part which is sensible. If these are indeed the Hamiltonians used for the simulations, I would put a big question mark on the conclusions presented in Figs. 3- 5. This is implied in the figure labeling but of course I cannot judge what was used in the simulations.

We did not try to convince that the model Hamiltonians, Eq. (2.2b-c) could show which part of the full high field truncated Hamiltonian "was responsible for the polarization transfer under different

conditions". The full Hamiltonian was used for simulations, unless noted. The goal of the model Hamiltonians was mentioned on the page 7, lines 152-154: "*Firstly we investigate which part of the full high field dipolar Hamiltonian can be a model Hamiltonian. A model Hamiltonian is a simplified Hamiltonian, which provides the same evolution of the spin system at specific time points as a full dipolar Hamiltonian."* .

However, we agree that a sentence in the end of the previous section was unclear (page 7, lines 137-138): "*We investigate the influence of each part of the Dipolar Hamiltonian (the secular and the flip-flop parts) on the measured operators at specific time points under different simulated conditions."* We apologize for it. We modified this sentence (page 7, lines 137-138 in the revised article): "*We investigate the behavior of the measured operators at specific time points under different simulated conditions, by separately analyzing different parts of the full high field truncated Dipolar Hamiltonian (the secular and the flip-flop parts).*"

Eqs.(2.2a-c) (page 7, lines 155-156) represent the model Hamiltonians. Using the model Hamiltonian we cannot make any conclusions about the model Hamiltonian itself, but the evolutions of the operators at specific points. Under conditions when the differences between offset values of the spins significantly smaller with respect to the used MAS rate, the model Hamiltonian (2.2a) provides the same evolutions of the operators as with the full high field truncated Hamiltonian (Eq. 2.1). It is shown on Figs. (2a and 4a). Under conditions when the differences between offset values of the spins is not small with respect to the used MAS rate, the model Hamiltonian (Eq. 2.2c) provides the same evolutions of the operators at specific time points as with the full high field truncated Hamiltonian (Eq. 2.1), which is shown in Fig. 3b. The model Hamiltonian Eq. (2.2b) was used as a part of the investigation.

On page 11, the heteronuclear Hamiltonian is discussed: "The main difference to the heteronuclear full dipolar Hamiltonian with a homonuclear model Hamiltonian (Eq. 2.1a) is a factor of 1.5." This is clearly not correct. The heteronuclear Hamiltonian has just the zz terms and is exactly the same as a homonuclear (weakcoupling) dipolar Hamiltonian. There is no factor of 1.5 difference between the two. Where we can find a factor of 1.5 is in the transition frequencies after diagonalisation. All these statements that are clearly wrong make me worry about the results presented in this paper.

A model Hamiltonian is a simplified Hamiltonian, which provides the same evolution of the spin system at specific time points as a full dipolar Hamiltonian. Eq. (2.2a) has the same structure as a full high field truncated heteronuclear dipolar Hamiltonian. The difference is a factor 1.5. In our simulations when we compared the evolution of the homonuclear and heteronuclear spin systems, we kept the ratio between homonuclear and the heteronuclear dipolar coupling values as 1:1.5 (Figs. 5-7). Under this condition and the negligible small difference of the offset values of the spins the heteronuclear and homonuclear fp-RFDR signals completely coincide at whole number of rotor periods. We used the model Hamiltonians only to make Figs. 2-4. For all other Figs. the full dipolar Hamiltonian was applied. We will add the sentence about it. We also will add to the captions of Figs. 5-9 that: "*The full dipolar Hamiltonian (Eq. 2.1) was used to evaluate the $I_2$ spin system and the secular Hamiltonian was used to evaluate IS spin system.*"

There is another statement that worries me (page 13): "For negligible offset differences with respect to the MAS rate, the evolution of the operators of IS and I2 spin systems are the same at specific time points." What are negligible offset differences in a heteronuclear spin system when the two spins are in different rotating frames? And further on: "With increase of offset difference the IS spin system passes through specific rotor resonance condition (the difference between offsets equal to half of the MAS rate), under which the transfer does not occur. For the I2 spin system the velocity of the transfer increases with increased offset difference." This cannot be. Either the authors mix up homo- and

heteronuclear spin systems or this is plain wrong. There are no MAS resonance conditions in heteronuclear spin systems that depend on any offsets (what offset differences are there?).

For $I_2$ spin system the applied internal Hamiltonian was:
$$H = \Omega_{I1}I_{z1} + \Omega_{I2}I_{z2} + 0.5\omega_{D,II}(t)[3I_{z1}I_{z2} - \bar{I}_1\bar{I}_2], \quad \text{Eq. (R1)}$$
whereas the heteronuclear internal Hamiltonian was:
$$H = \Omega_I I_z + \Omega_S S_z + \omega_{D,IS}(t)I_z S_z. \quad \text{Eq. (R2)}$$
The offset difference was meant as $|\Omega_I - \Omega_S|$ for heteronuclear and $|\Omega_{I1} - \Omega_{I2}|$ homonuclear spin systems. Although an offset difference for the heteronuclear case may seem irrelevant when each spin is in a different rotating frame (as was noticed by reviewer), it can be seen from Eqns R1-2 that the offsets appear with the same mathematical form. The appearance of a resonance condition is perhaps less surprising when considering that essentially identical RF pulsing is applied on both channels by design of the pulse sequence. Fig. 5c (page 14) shows the case when the transfer between the operator $I_z$ and $S_z$ has negligible small value. It occurs when $|\Omega_I - \Omega_S| = 0.5\omega_R$ and the heteronuclear dipolar coupling is smaller with respect to MAS rate. Since we are first who show the heteronuclear fp-RFDR transfer, such "rotor resonance condition" has never been mentioned. The same results and conclusions can be obtained with SIMSPON simulations. Some SIMPSON codes for $I_2$ and *IS* spin systems under different simulated conditions are shown in the end of this response.

Even if some of the problems are just a mix up, all of these mistakes make me wonder how carefull the rest of the work is done. I have not and will not look at the rest of the manuscript before all these problems are corrected or explained.

We hope that after these clarifications, the reviewer will continue to review our article. This manuscript has been prepared with a great deal of care, and simulation results have been checked against SIMPSON.

Some minor remarks that should be cleaned up:

(1) The introduction has a long list of references to homonuclear and heteronuclear recoupling experiments that is clearly incomplete. Instead of such an incomplete list, it would be much better to have a reference to a review. There is for example a recent review about dipolar resoupling by N.C. Nielsen, L.A. Strassø, A.B. Nielsen, (Top. Curr. Chem. 306 (2012) 1–45) as well as older reviews on his topic. I think it would be more appropriate as a reference than an incomplete list.

We will include that review into citation. We have made no attempt to provide a complete list of recoupling experiments and hope the reviewer can forgive omission of any particularly relevant literature.

2) page 5 line 99/100: Liovuille-von Neumann equation is much more common. Why hbar=1? Usually the Hamiltonian in NMR is in frequency units and then there is no hbar in front of the commutator.

We removed $\hbar$ and mentioned Eq. (1.1) as an equation of motion.

(3) Equation numbering should be continuous from (1) to (N) without chapter subdivisions.

We will modify all equation numbering according to this suggestion.

We prepared three different SIMPSON codes, representing $I_2$ (code 1) and *IS* (codes 2 and 3) fp-RFDR sequences. The transferred signals with codes 1 and 2 coincide completely, since all spins are onresonance and the ratio between the heteronuclear and homonuclear dipolar coupling constants is 1.5. The code 3 represents the case, when the difference between offsets of the heteronuclear spins is half of MAS rate and the dipolar coupling constant is smaller with respect to MAS rate. The transferred fp-RFDR signal is negligibly small.

**Code 1**

```
spinsys {
 channels 1H
 nuclei   1H 1H
 dipole 1 2 -2e3 0 0 0
}

par {
 proton_frequency 600e6
 spin_rate      10000
 sw             spin_rate/8
 np             20
 crystal_file   rep100
 gamma_angles   14
 start_operator  I1z
 detect_operator  I2z
 verbose        1101
 variable rfq    50000
}
proc pulseq {} {
 global par

 maxdt 1

 set t180 [expr 0.5e6/$par(rfq)]
 set DELAY [expr (1e6/$par(spin_rate))-$t180]

 reset
 pulse $t180 $par(rfq) x
 delay $DELAY
 pulse $t180 $par(rfq) y
 delay $DELAY
 pulse $t180 $par(rfq) x
 delay $DELAY
 pulse $t180 $par(rfq) y
 delay $DELAY
 pulse $t180 $par(rfq) y
 delay $DELAY
 pulse $t180 $par(rfq) x
 delay $DELAY
 pulse $t180 $par(rfq) y
 delay $DELAY
 pulse $t180 $par(rfq) x
 delay $DELAY

 store 1
```

```
    reset
    acq
    reset

    for {set i 1} {$i <$par(np)} {incr i} {
    prop 1
    acq
    }
}

proc main {} {
    global par

    set    f [fsimpson]
    fsave $f $par(name).fid
    fsave $f $par(name).dat -xreim
}
```

**Code 2**          %%%%%%%%%%%%%%%%%%%%%%%%%%%%%%%%%%%%

```
spinsys {
    channels 1H 13C
    nuclei   1H 13C
    dipole 1 2 -3e3 0 0 0
    shift  1 0e3 0 0 0 0 0
    shift  2 0e3 0 0 0 0 0
}

par {
    proton_frequency 600e6
    spin_rate       10000
    sw              spin_rate/8
    np              20
    crystal_file    rep100
    gamma_angles    14
    start_operator  I1z
    detect_operator I2z
    verbose         1101
    variable rfq    50000
}
proc pulseq {} {
    global par

    maxdt 1

    set t180 [expr 0.5e6/$par(rfq)]
    set DELAY [expr (1e6/$par(spin_rate))-$t180]

    reset
    pulse $t180 $par(rfq) x   $par(rfq) x
    delay $DELAY
    pulse $t180 $par(rfq) y  $par(rfq) y
    delay $DELAY
    pulse $t180 $par(rfq) x   $par(rfq) x
    delay $DELAY
    pulse $t180 $par(rfq) y  $par(rfq) y
```

```
  delay $DELAY
  pulse $t180 $par(rfq) y  $par(rfq) y
  delay $DELAY
  pulse $t180 $par(rfq) x  $par(rfq) x
  delay $DELAY
  pulse $t180 $par(rfq) y  $par(rfq) y
  delay $DELAY
  pulse $t180 $par(rfq) x  $par(rfq) x
  delay $DELAY

  store 1
  reset
  acq
  reset

  for {set i 1} {$i <$par(np)} {incr i} {
  prop 1
  acq
  }
}

proc main {} {
  global par

  set   f [fsimpson]
  fsave $f $par(name).fid
  fsave $f $par(name).dat -xreim
}
```

**Code 3**      %%%%%%%%%%%%%%%%%%%%%%%%%%%%%%%%%%

```
spinsys {
  channels 1H 13C
  nuclei   1H 13C
  dipole 1 2 -3e3 0 0 0
  shift  1 -2e3 0 0 0 0 0
  shift  2 3e3 0 0 0 0 0
}

par {
  proton_frequency 600e6
  spin_rate       10000
  sw              spin_rate/8
  np              20
  crystal_file    rep100
  gamma_angles    14
  start_operator  I1z
  detect_operator I2z
  verbose         1101
  variable rfq    50000
}
proc pulseq {} {
  global par

  maxdt 1
```

```
set t180 [expr 0.5e6/$par(rfq)]
set DELAY [expr (1e6/$par(spin_rate))-$t180]

reset
pulse $t180 $par(rfq) x   $par(rfq) x
delay $DELAY
pulse $t180 $par(rfq) y  $par(rfq) y
delay $DELAY
pulse $t180 $par(rfq) x   $par(rfq) x
delay $DELAY
pulse $t180 $par(rfq) y  $par(rfq) y
delay $DELAY
pulse $t180 $par(rfq) y   $par(rfq) y
delay $DELAY
pulse $t180 $par(rfq) x  $par(rfq) x
delay $DELAY
pulse $t180 $par(rfq) y   $par(rfq) y
delay $DELAY
pulse $t180 $par(rfq) x  $par(rfq) x
delay $DELAY

store 1
reset
acq
reset

for {set i 1} {$i <$par(np)} {incr i} {
prop 1
acq
 }
}

proc main {} {
 global par

 set   f [fsimpson]
 fsave $f $par(name).fid
 fsave $f $par(name).dat -xreim
}
```

1. Vega, S. Fictitious spin 1/2 operator formalism for multiple quantum NMR. *J. Chem. Phys.* **68**, 5518–5527

   (1978).

2. Bennett, A. E., Griffin, R. G., Ok, J. H. & Vega, S. Chemical shift correlation spectroscopy in rotating solids:

   Radio frequency-driven dipolar recoupling and longitudinal exchange. *J. Chem. Phys.* **96**, 8624–8627 (1992).

3. Ishii, Y. 13C–13C dipolar recoupling under very fast magic angle spinning in solid-state nuclear magnetic resonance: Applications to distance measurements, spectral assignments, and high-throughput secondary-structure determination. *J. Chem. Phys.* **114**, 8473–8483 (2001).

4. Nimerovsky, E. & Goldbourt, A. Insights into the spin dynamics of a large anisotropy spin subjected to long-pulse irradiation under a modified REDOR experiment. *J. Magn. Reson.* **225**, 130–41 (2012).

---

## Referee Comment (RC2) · Anonymous Referee #1 · 8 Dec 2020

I would like to thank the authors for the extensive explanations and response to my original review. While it clarifies some points, there are still points where I disagree with the paper and explanation. I will respond to some of the comments that I consider still unclear and not solved.

(1) The authors talk about offsets and chemical-shift differences. While I understand that in the homonuclear case the chemical-shift difference plays an important role in the classical description of RFDR, I do not see that the chemical-shift difference is important in the heteronuclear case. I can see that offset effects are important because they change the nutation frequency and the nutation axes and introduce small errors in the 180 degree pulses that could play an important role in the polarization transfer. This might sound like semantics but I think this is an important point that needs to be
clarified whether offsets or chemical-shift differences are relevant in which case.

(2) In the Theory section, the authors talk about analytical integration of the Hamiltonian as written in Eq. (1.4) that is used in the numerical simulations. I still think that in the context discussed here, Eq. (1.4) is wrong and needs a Dyson time ordering operator in front of the exp{ } term. The Hamiltonian used here cannot be integrated analytically because it does not commute with itself at different times even over a small time periode. The only exception is the heteronuclear case without rf pulses where the Hamiltonian is diagonal in the product basis and can be integrated analytically. In all other cases, either rf on both channels or homonuclear dipolar couplings, there are no analytical solutions. Of course, one can do the integration in a first-order average Hamiltonian sense but then this is not an analytical solution. This can also be seen from Fig. 1 where the simulation changes with increasing N which would not be the case if the integration could indeed be solved analytically. Again, this could be semantics, but I think it is important semantics because an analytical solution is clearly defined.

(3) Model Hamiltonians: Of course, one can use any model Hamiltonian that one wishes to use and see whether it can explain the polarization transfer pathways observed in the experiments. However, it would be good if these model Hamiltonian make also physical sense. I can understand that one argues that the transverse terms in the Hamiltonian are truncated leaving only the zz terms but I cannot understand why on e would use a truncated Hamiltonian scaled by 3/2 for the homonuclear case. I also do not see why one would use an isotropic Hamiltonian because if the zz term is not relevant, the zq Hamiltonian would remain. I think it is important to motivate why certain model Hamiltonians are considered.

I had a quick look at the experimental section and was wondering why the conditions are different from most of the simulations. Especially in Table 3 and 4 it says 100 kHz MAS and rf fields of 50.2/47/49.31 kHz. How can one get a 180 degree pulse in a rotor period if the rf field is smaller than half the spinning frequency? Is the description
presented in the theory and simulations still correct for these conditions? This is basically a windowless pulse sequence (with some of the pulses obviously truncated short) and is very close to the CPPI method of Reif (without the amplitude jump). In addition, this is basically the n=0 Hartmann-Hahn condition where second-order transfer could happen. Have the authors evaluated whether such contributions are a source of the experimental polarization transfer? These conditions don't seem to be part of the simulations that are done mostly at much slower MAS rates with a much lower ratio of pulse and rotor period. I start wondering whether the experiments are not better described by two synchronized R sequences on both channels. As far as I can remember there are some papers about such sequences.

All in all, I think there are too many (sometimes minor) problems that add up and make a judgement of the paper as a whole very difficult.

I think this paper needs a major rewrite to make it more consistent and have a better connection between the parts. I would eliminate the theory/simulation part and start around Fig. (5) to illustrate the different pathways and then make sure that simulations and experimental conditions match.

MRD

---

## Referee Comment (RC3) · Anonymous Referee #2 · 10 Dec 2020

The manuscript by Evgeny et.al proposes the use of RFDR for heteronuclear polarization transfer in the fast MAS regime. Instead of the routinely used time-independent effective Hamiltonians to describe recoupling conditions, the authors use the approach of time slicing typically employed in numerical simulation to follow the fate of different operators during heteronuclear fp-RFDR. They have explored several conditions for polarization transfer in presence and absence of CSA, offsets etc. Some experiments have been performed in the end to demonstrate the feasibility of heteronuclear polarization transfer. The experimental comparison demonstrates inferior performance than ramped-cp. I think using fp-RFDR for heteronuclear polarization transfer is an interesting idea and could potentially be beneficial in some special circumstances.

Having said the above I think there are several aspects that need to be carefully considered and explained before the potential of the method can be fuly appreciated.

The proposal of heteronuclear RFDR recoupling gets lost in the theoretical description and simulations under different conditions, model Hamiltonians, homo and heteronuclear cases. I think the section is too long and repeatedly the same message is conveyed through first 4-5 figures. From this description it is difficult to follow how the heteronuclear recoupling depends on factors such as the offset, duty factor etc as they seem very relevant in defining the recoupling condition/conditions. Both the role of offsets and duty factor is very crucial in defining the polarization transfer efficiency. I think the authors also realize this therefore the repeated one-D experimental data at different offset conditions. As is generally the norm, it would be useful to show the effective Hamiltonians (with relevant parameters) and then differentiate the conditions of homonuclear and heteronuclear polarization transfer. Lines 211-221: . . . . . . the first conclusion diverges – dependence on the ZQ Hamiltonian . . .. For AHT. I think the authors attempt to differentiate the homonuclear and heteronuclear polarization transfer conditions. This is not very clear and the authors should highlight it explicitly along with the aspects discussed above.

In principle one could follow the behaviour of the different operators in SIMPSON by setting the detection operator to the desired coherence. So I do not see the additional insights that one gains from the theoretical time slicing approach.

In Eq.4, I am wondering whether one can really get rid of the Dyson time ordering operator. I do not even see a clearly defined initial Hamiltonian. Before doing anything it is absolutely necessary to define internal interaction and rf Hamiltonian and systemically walk the reader through the analysis.

The idea of heteronuclear RFDR is not new and was proposed almost 25 years ago by Griffin and co-workers albeit at lower MAS frequency. (JMR A, 112, 191-198)

In my opinion, the presented experimental data is completely disconnected from the theoretical description and simulation provided in the beginning of the article.

[Figure]

How does a 180 pulse applied with 47 kHz amplitude fit into 10 microsecond rotor period? Similarly, a 50 kHz 180-degree pulse on 1H with a 10-microsecond rotor period would imply a continuous pulsed rather than finite pulsed rfdr. are these errors necessary for experimental polarization transfer.

During heteronuclear RFDR, the homonuclear polarization transfer amongst proton and carbons is still operational. Its not clear how this simultaneous homonuclear polarization transfer impacts the heteronuclear polarization. There is also no discussion on this aspect. This is relevant in context of the conclusion drawn in line 480-481.

Eq 2.1, I think one 0.5 factor is extra.

The physical basis of model Hamiltonians is unclear. Also the condition where the particular model Hamiltonians will become relevant should be clearly highlighted.

Line 195-200: The authors compare buildup up of polarization due to IzSz Hamiltonian at slow and fast MAS. The conclusion that buildup rate is reversed is vague. I think the rate of polarization transfer critically dependence on the number of pulses applied. Since at 90 kHz MAS more pulses have been applied within 1-3 ms therefore the polarization builds up faster. I am bit surprised that the authors continuously discuss recoupling during the pulses in the manuscript but do not take this into account while discussing differential build up rates. In this light figure 2-4, needs a bit more careful analysis in terms of offsets and number of pules applied during the recoupling period.

The evolution of magnetization during finite pulse will lead to loss of magnetization so unlike other heteronuclear polarization transfer sequences such as CP where magnetization reach an equilibrium around 50%. Here one would expect the magnetization to rapidly decay. This is already visible in the appendix figures. However it would be good if one could show this both in experiments and simulation in a more realistic spin system. I think even in the ideal IS spin system the magnetization would decay after some finite number of pulses. It is very important to distinguish this aspect from regular heteronucler polarization methods.

Line 265-266: In case of the IS spin system, when the sum of the two offsets is equal to half the MAS frequency how can we understand the loss of magnetization. A similar phenomenon is not observed in case of homonuclear spin system. Its not clear why and what leads to the loss of polarization?

Figure 6: I am bit at a loss to understand what is it that differentiates figure 6a and 6b except the label IS and I2. For both the spin systems, offset is zero, csa's are zero and the same rf field is used i.e the rotating frame is the same. So why do differences occur. I am assuming the full Hamiltonian is used for the simulation since nothing is explicitly mentioned. Why do amplitudes of different operators change for the homonuclear case but not for the heteronuclear case.

In all heteronuclear simulations the same rf field is used on the two nuclei. Of course this is more challenging experimentally. It would be good to depict a plot of polarization transfer efficiency as a mismatch of the two rf amplitudes. This would also depict how forgiving the method is to rf missets and should be compared to a similar CP profile under identical conditions.

Figure 9: The transfer is $I1z \rightarrow I2z$ i.e homonuclear but heteronuclear spin system parameters are used. Is this correct or it does not matter, not clear?

It is claimed that polarization transfer to aromatic group during hetRFDR is comparable to CP. However it is not clear whether rCP was optimized for broadband polarization transfer or is it compared directly to aromatic peak optimized rCP. It would also help to have experimental polarization buildup curves as a function of mixing time for both transfer.

The low S/N noise due to 13C/15N detection obscures a meaningful comparison between hetRFDR and CP transfer. It might be meaningful to use two hetRFDR to do proton detection and clearly have the sensitivity to highlight the gain on different chemical moieties.

Minor comments a) line 183 . . ..pink lines . . . . . . pink and magenta are interchangeably used in figure caption b) several figure captions eq 2.2c is referred as scalar Hamiltonian. c) Several instances of use of . . ... "Microscopic and macroscopic" magnetization −→ I think the author try to imply single crystal and powder averaging. Its better to use the later to be consistent with literature unless something more is implied. d) Several instances: velocity of magnetization transfer is used -→ one should consider using polarization transfer or buildup rates. e) Line 438: reference spell check.

I feel there are number of issues with the manuscript. Theory, simulations and experiments do not seem to provide a meaningful perspective for the proposed heteronuclear RFDR recoupling. Authors should really reconsider streamlining the theory and experiments to make the manuscript appealing.

———————————————————

---

## Referee Comment (RC4) · Anonymous Referee #3 · 4 Jan 2021

This manuscript presents an operator analysis of the well-known fp-RFDR recoupling sequence with some emphasis on the XY phase cycling scheme. The benefit of this analysis should have been defined at the end of the introduction.

Allow me to make some comments on the text:

115: What is meant here by an analytical integration?

120,146: "by I3 spin system". Please define this system- "by THE/A I3 spin system".

130: what is the definition of f in fïĄőCSA,2.

147,155: sum over r<s=1,2,3 ? and add a dot between Ir and Is.

152: If the model Hamiltonian provides an equivalent spin evolution as of the full Hamil-

tonian at specific time points, then there are two possibilities. If the specific points are equally spaced in time, I would call it an average Hamiltonian. If they are not equally spaced in time, then I do not see the purpose of introducing a "model" Hamiltonian. I can appreciate the fact that the average Hamiltonian can be derived not theoretically but numerically, but then it should be general enough, in terms of variations of the spin parameters, to be significant.

164: That the red and black lines cross each other is not a surprise. If the amplitudes of all operators would cross the corresponding black operators at the same equally spaced positions then we can think about an average Hamiltonian.

— 194: Would the result at this point mean that the three model Hamiltonians in Eq. (2) are not adequate to provide a general form of the average Hamiltonian?

213: That the secular Hamiltonian plays a significant role in the transfer is not really a surprise. Without this part of the Hamiltonian there would be no dipolar interaction. The flip-flop terms can be (partially) quenched because of off-resonance effects, thus the conclusions on the top of page 11 are surprising. They are however not very practical when dealing with CSA tensors in rotating samples where the frequency differences are modulated by the spinning. Therefore, there seems not to be a clear conclusion about the choice of the three model Hamiltonians and in the rest of the manuscript, the full Hamiltonian is applied in all cases.

When dealing with the IS spin system it should be stated from the beginning that the fp-RFDR sequence looks like Fig. 10.

224: The "same" value of the dipolar interaction strength results for a homo-nuclear spin pair in a static spectral line splitting that is 50% larger than for a hetero-nuclear spin pair. Thus, only when spectral frequencies are considered and for "display reasons" their values are compared, it is practical to increase the hetero-interaction artificially by a factor of 1.5. In MAS one should be careful doing so, because the sideband patterns are not straightforwardly showing this 1.5 spectral factor.

238–: What is the purpose of the extended analysis of all 14 operators as a function of offset. What do they tell us? Here again it would have been nice to know what the aim of this part of the study is. From the experimental point of view, what is of interest is the operator that results in the signal. That particular operator should have been emphasized. The study does not tell us about the yield of polarization transfer. Following the text following page 11, much details are presented and it is hard to follow the main line of thought. Perhaps summarizing the conclusions, followed by some examples (and moving part of the figures to a supplementary file), would help the reader to comprehend what is going on. Also the effects/benefits of the XY phase cycling should be characterized separately from the corent pathways.

416 –: If part of the numerical derivations are intended to introduce the yield of the hetero-nuclear polarization transfer experiments, then that should have been stated from the beginning. Here again the various comparisons between the experimental spectra are a bit confusing. Perhaps some consistent conclusions at the end of page 22 can form a basis for the analysis of the spectra.